# The Effects of Processing on Bioactive Compounds and Biological Activities of Sorghum Grains

**DOI:** 10.3390/molecules27103246

**Published:** 2022-05-19

**Authors:** Zhenhua Li, Xiaoyan Zhao, Xiaowei Zhang, Hongkai Liu

**Affiliations:** 1College of Agriculture, Guizhou University, Huaxi District, Guiyang 550025, China; 2Department of Food Science and Nutrition, College of Culture and Tourism, University of Jinan, Jinan 250002, China; st_zhaoxy@ujn.edu.cn (X.Z.); st_zhangxw@ujn.edu.cn (X.Z.)

**Keywords:** sorghum, decortication, thermal process, germination, enzymatic treatment

## Abstract

Sorghum is ranked the fifth most commonly used cereal and is rich in many kinds of bioactive compounds. Food processing can affect the accumulation and decomposition of bioactive compounds in sorghum grains, and then change the biological activities of sorghum grains. The present review aims to analyze the effects of processing technologies on bioactive compounds and the biological activities of sorghum grains. Decortication reduces the total phenols, tannins, and antioxidant activity of sorghum grains. The effects of thermal processes on bioactive compounds and potential biological activities of sorghum grains are complicated due to thermal treatment method and thermal treatment conditions, such as extrusion cooking, which has different effects on the bioactive compounds and antioxidant capacity of sorghum due to extrusion conditions, such as temperature and moisture, and food matrices, such as whole grain and bran. Emerging thermal processes, such as microwave heating and high-pressure processing, could promote the release of bound phenolic substances and procyanidins, and are recommended. Biological processes can increase the nutritive and nutraceutical quality and reduce antinutritional compounds, except for soaking which reduces water-soluble compounds in sorghum.

## 1. Introduction

Sorghum (*Sorghum bicolor* L. Moench) is ranked the fifth most commonly used cereal worldwide after wheat, rice, maize, and barley [1,2,3]. The essential components of sorghum grain are starch, fat, protein, and non-starch polysaccharides [4]. It also contains a wide variety of bioactive compounds, such as vitamin E, carotenoids, and phenolic compounds [4,5,6]. Sorghum has been considered a source of nutraceutical and functional nutrients. Multiple in vitro and in vivo studies have shown that sorghum grain has many biological activities, including antioxidation, anti-inflammation, antithrombotic and antidiabetic properties [2,3,7,8].

However, some bioactive compounds are very sensitive when exposed to light, heat, and oxygen. Food processing includes mechanical treatments, traditional thermal treatments, non-thermal treatments, and biological treatments (Figure 1). It is the conventional treatment of cereal before human consumption and may modify the functional and nutritional properties of bioactive compounds in sorghum grains [9,10]. Before consumption or processing to make them palatable, sorghum grains generally undergo all kinds of processing. Throughout the years, decortication, milling, soaking, steaming, boiling, pressure cooking, roasting, oven, microwave heating, extrusion, germination, fermentation, and enzymatic treatments have been used in the sorghum grains processing to meet the demands of the consumer. Some processing technologies could decrease the functional and nutritional properties of bioactive compounds. Furthermore, the sorghum grain is formed by the pericarp, the testa, the endosperm, and the germ [9]. The pericarp and the testa, which are rich in carotenoids, anthocyanins, phenolic acids, flavonoids, and tannins, are often removed for taste, causing the decrease of these bioactive compounds. Some processing technologies like germination, fermentation, and enzymolysis could increase the content of bioactive compounds or even synthesize new bioactive compounds [11,12]. Therefore, how the processing technologies change the bioactive compounds and the effective utilization or improvement of these processing technologies to maintain or increase the bioactive compounds need to be studied.

Currently, several informative reviews have depicted the bioactive compounds and biological activities in sorghum grains [13,14,15,16], whereas there is a paucity of reviews documenting the effect of kinds of processing technologies on bioactive compounds and biological activities in sorghum grains. Hence, the present review aims to analyze the effects of processing technologies on bioactive compounds and the biological activities of sorghum grains.

## 2. Bioactive Compounds in Sorghum

Sorghum contains various bioactive compounds that are secondary plant metabolites [15]. The bioactive compounds are primarily the phenolic compounds in sorghum. The dominant polyphenols in sorghum are structurally classified as phenolic acid, flavonoid, procyanidin, and stilbenoids [4,6]. Phenolic acids are the most important phenolic compounds and have thus been widely studied. The major phenolic acids identified in sorghum grain include caffeic acid, p-coumaric acid, ferulic acid, sinapic acid, chlorogenic acid, protocatechuic acid, p-hydroxybenzoic acid, vanillic acid, salicylic acid, gallic acid, and syringic acid (Figure 2A,B). Flavonoids are mainly found in the outer layers of the grain, and the types, concentration, and profile of flavonoids are associated with the pericarp color and thickness [6,17]. Flavones, flavanones, flavonols, flavanols, dihydroflavonol, anthocyanins, and isoflavones have been previously identified in sorghum. Anthocyanins account for 79% of the flavonoids’ content and belong to the class of 3-deoxyanthocyanidins [17]. Notable, the 3-deoxyanthocyanins are almost uniquely found in sorghum and rarely in other food plants [3,16]. Moreover, 3-DXAs are stable in solution compared to other anthocyanidins. The main 3-deoxyanthocyanidins of the sorghum are nonmethoxylated forms are luteolinidin, apigeninidin, 5-methoxyluteolinidin, 7-methoxyapigeninidin, 5-methoxyluteolinidin, 7-glucoside, luteolinidin 5-glucoside, 7-methoxyapigeninidin 5-glucoside, apigeninidin 5-glucoside, and luteolinidin anthocyanin (Figure 2C). Moreover, condensed tannins in sorghum (Figure 2D), which have higher levels and molecular weight than other cereal grain tannins, are a special class of phenolic compounds [6]. In addition, carotenoids, vitamin E, amines, policosanols, and phytosterols were also reported in sorghum [3].

## 3. The Change of Bioactive Compounds and Biological Activities of Sorghum during Mechanical Treatments

### 3.1. Decortication

Decortication is the process of removing the grain’s pericarp and most of the testa layer. First, it can eliminate the bad taste the bran brings. Second, some antinutrients are removed, such as tannins, resulting in an increasing the bioavailability of proteins [19]. Moreover, it can be in favor of the storage of sorghum products because the oxidation of lipophilic compounds in bran generally leads to the development of rancid off-odors and flavors [20]. However, decortication reduced the total phenols, tannins, and antioxidant activity of sorghum grains (Table 1), because the polyphenols are concentrated in the grain’s pericarp and testa layer. Furthermore, these antinutrients, such as tannins, are proved to prosses many health benefits. Therefore, a whole sorghum grain diet should be advocated and new technologies should be developed to solve the problem of off-odors and taste.

### 3.2. Soaking

Sorghum grains often are soaked before further processing. Soaking for a certain time can lead to the physical and chemical changes of seeds. During soaking, water enters the kernel, leads to rupture of the grain cells, swells and softens the physical structure, and releases the water-soluble compounds [21,22]. Some studies showed soaking significantly decreased total phenols, total flavonoids, tannins, phenolic acids compounds, flavonoid components, vitamin E, β-carotene, antioxidant activity, α-glucosidase, and α-amylase inhibitory activities of sorghum grains [22,23,24]. However, Xiong et al. (2019) showed that soaking increased the total flavonoid content in sorghum, while total phenolic content, condensed tannin content, and antioxidant activity were not affected; this may be related to the slight increase of total phenolic content, which might be due to the release of some bound phenolic compounds and the minor loss of total phenolic content, and condensed tannin content might be too negligible to cause statistically significant differences [21]. In brief, soaking can reduce water-soluble compounds in sorghum.

Moreover, some studies have reported that the improved mineral bioavailability and digestibility by reducing anti-nutrients after soaking in other seeds of cereal crops [19]. There has been no report on the effect of soaking on mineral bioavailability and protein digestibility. More studies are needed to analyze the effect of soaking on nutrient composition, the utilization of sorghum food digestibility, and absorption.

## 4. The Change of Bioactive Compounds and Biological Activities of Sorghum during Thermal Processes

Sorghum grains are often subjected to various forms of thermal processing. The thermal process can successfully inactivate microbes and enzymes in sorghum to ensure food safety, but it also has a negative impact both on the nutritional and sensory properties of sorghum products. The traditional thermal process can be divided into wet cooking (including steaming, boiling, etc.) and dry cooking (including roasting, etc.). Microwave heating and high-pressure boiling are highly popular in a variety of industrial and domestic applications, and extrusion is also widely applied in the food industry.

Generally, wet cooking decreases the contents of bioactive compounds and potential biological activities, and may increase the contents of some free phenolic compounds (Table 1). For example, the steaming process (100 °C for 50 min) greatly reduced (*p* ≤ 0.5) the TPC, TFC, and CTC in black sorghum [24], and the tannin content decreased significantly during cooking (17%) and steaming (35%) [25]. That may be due to the degradation of the antioxidants at higher temperatures, the damage to the cellular structure of grains causing the release of some bound phenolic compounds, and the dissolution of water-soluble compounds into the water or steam [21,25].

As seen in Table 1, roasting increased the phenolic contents of sorghum. The possible cause might be the breakdown of cellular constituents and cell walls causing the release of bound phenolic acids, the degradation of some conjugated polyphenolics into simple phenolics, and the increased extractability of some phenolic components [21,23]. Cardoso et al. (2014, 2015) showed that dry heat in the conventional oven increased the vitamin E, and decreased the carotenoids, flavanones, flavones, and proanthocyanidins of the sorghum flours. However, it did not affect 3-deoxyanthocyanidins. Additionally, the antioxidant activity in processed flours with dry heat in the conventional oven remained constant or increased. That may be because various bioactive substances have different sensitivity to heat and the carotenoids, flavanones, flavones, and proanthocyanidins are more sensitive than vitamin E to heat.

Extrusion cooking is an industrial cooking process that combines high pressure, heat, mechanical shear, and low moisture by producing structural alterations and changes in functional properties in a short period [26,27]. Some studies showed that extrusion cooking decreases the bioactive compounds and antioxidant capacity of sorghum, such as phenolic compounds, and carotenoids (Table 1). Lopez et al. (2016) revealed that the extrusion process increased total phenol content in sorghum bran compared to non-extruded sorghum, particularly for extrusion at 180 °C with 20% moisture content microwave heating. That may be because heat treatment of cereals enhanced the release of phenolic acids and their products from the cell walls. Therefore, the effects of extrusion cooking on bioactive compounds in sorghum may depend on the extrusion condition such as temperature and moisture and food matrices such as whole grain and bran. The detailed mechanisms thereof need more research.

Microwave heating and high-pressure processing are two new alternatives to traditional heat treatments and possess some benefits such as energy saving, high efficiency, and less damage to nutrients [28]. Their influences on bioactive compounds are different according to current research. They could promote the release of bound phenolic substances and procyanidins from the results of the studies. The effects and detailed mechanisms need more research.

To sum up, thermal processes may reduce some sensitive bioactive compounds of sorghum, but they can promote the release of bound phenolic compounds and antinutrients. Therefore, thermal processes can be recommended for sorghum products containing bran.

## 5. The Change of Bioactive Compounds and Biological Activities of Sorghum during Biological Processes

### 5.1. Germination/Spouting

Germination, also known as sprouting, is a process of beginning with water absorption by inactive grains and ending with the emergence of the embryonic axis. It brings the degradation of main macronutrients, reduction of anti-nutritional factors, and increment of different bioactive compounds and many bioactivities [29]. It is an inexpensive and effective method for increasing the nutritive and nutraceutical quality of cereal and legume seeds. As seen in Table 1, most studies show that germination increases phenolic compound levels and antioxidant activity of sorghum sprouts. Hithamani and Srinivasan (2014) found that germination did not cause any significant change in the total phenol content and significantly reduced total flavonoid content [30]. The principal reason is that the synthesis of bioactive compounds in sprouts can be affected by germination conditions such as soaking time and temperature germination time and temperature, and elicitors such as light and ultrasound [29]. A study by Hassan et al. (2019) showed microwave processing of sorghum seeds could impart improvement in their composition regarding the phenolic profile and antioxidant activity of sorghum sprouts [10]. Hence, the higher contents of bioactive compounds need the optimization of sorghum germination conditions and even the help of elicitors. The products of the germinated sorghum should be developed.

### 5.2. Fermentation

Fermentation is a traditional food processing technique and has some advantages including improving digestibility, increasing nutritive value, and reducing anti-nutritional factors. Just like germination, the effects of fermentation on bioactive compounds depend on the fermentation conditions, such as the pH, fermentation time and temperature, and the kind and count of zymophyte. Table 1 shows that fermentation differently affected the bioactive compounds due to fermentation processing, but anti-nutritional factors were reduced during fermentation. Traditional Sudanese kisra is a natural lactic acid bacteria and yeast fermented sorghum thin pancake-like flatbread [31]. Ting is a sorghum fermented product, used for the preparation of gruel or porridge in Botswana, South Africa, and other neighboring countries [32,33]. Obtaining high values of total phenolic content and antioxidant activity of these sorghum products needs the optimization of fermentation conditions [31,32]. For example, at optimal fermentation conditions of whole-grain ting, high values of total phenolic content (46.1 mg GAE/g), total flavonoid content (40.9 mg CE/g), tannin content (14.1 mg CE/g), and antioxidant activity (3.7 µM TE/g) were obtained [32]. Lactic acid bacteria are dominant during the fermentation of sorghum in existing studies. During fermentation, the lactic acid bacteria can increase nutrients through microbial synthesis, food digestibility, and shelf life, and improve the palatability and acceptability by changes in flavor and texture [25,34]. Moreover, yeast was used in the production process of traditional Sudanese kisra [31]. Therefore, more fermentation microorganisms could be studied to develop more fermented sorghum products.

### 5.3. Enzymatic Treatment

The enzymatic treatment is an environmentally friendly technique. Recently, polyphenol oxidase, tannase, and phytas were used to reduce antinutritional compounds, such as tannins and phytate, in sorghum products [34,35]. Obviously, there are many enzymes which can be used in the treatment of sorghum. For example, some cell wall degrading enzymes could be used to break down the cell wall and release the phenolic compounds. Hence, more researches are expected to study the effects of kinds of enzymatic treatments on bioactive compounds and potential human health benefits.

## 6. Other Treatments

Nixtamalization is a traditional process in Mexico and Central America whereby corn is treated with lime, cooked, dried, and ground to produce the flour used to make tortilla. Gaytan-Martínez et al. (2017) studied the effect of the nixtamalization process on the content and composition of phenolic compounds and antioxidant activity of two sorghums. They concluded that nixtamalization effectively reduced condensed tannins to safety intake values, allowing the preservation of other phenolic compounds and antioxidant capacity [36].

Actually, the production of sorghum products may combine several treatments. The malting of sorghum includes the processes of steeping, germination, and drying [11], the production of traditional Sudanese kisra includes the processes of fermentation, baking, and drying [31], and obtaining sorghum grain tea requires the soaking, steaming, and roasting of raw sorghum grains [21,24]. Each treatment could influence the contents of bioactive compounds. Many effects of each treatment on bioactive compounds should be taken into account for developing new sorghum produce with a high content of bioactive compounds.

## 7. Conclusions

Food processing has significant effects on bioactive compounds and biological activities of sorghum grains. Decortication could reduce total phenols, tannins, and antioxidant activity of sorghum grains, wet cooking could decrease the contents of bioactive compounds and potential biological activities and may increase the contents of some free phenolic compounds, roasting increases the phenolic contents of sorghum, and extrusion cooking has different effects on the bioactive compounds and antioxidant capacity of sorghum due to the extrusion condition, such as temperature and moisture, and food matrix, such as whole grain and bran. Microwave heating and high-pressure processing could promote the release of bound phenolic substances and procyanidins. Soaking can reduce water soluble compounds in sorghum. Germination, fermentation, and enzymatic treatment can increase the nutritive and nutraceutical quality of sorghum.

The effects of many treatments on the bioactive compounds and biological activities of sorghum grains depend on the treatment conditions. Hence, the optimizing condition of various processing is important. Omics technologies, such as transcriptome, proteome, and metabolome, could be used to analyze the change mechanism of bioactive compounds. Sorghum products may combine several treatments; the methods combination and combined effects need to be studied further.

## Figures and Tables

**Figure 1 molecules-27-03246-f001:**
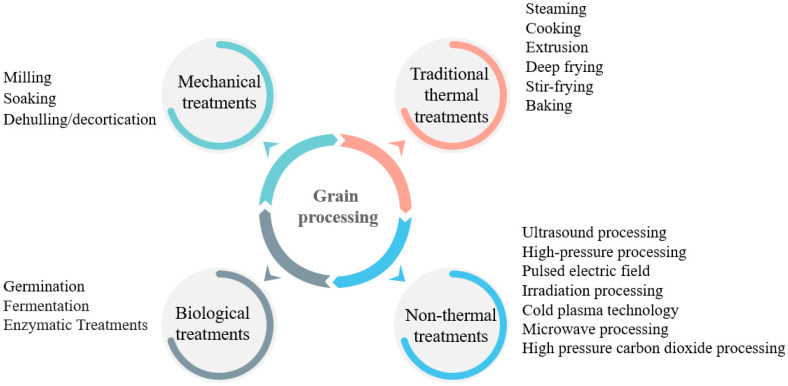
Grain processing.

**Figure 2 molecules-27-03246-f002:**
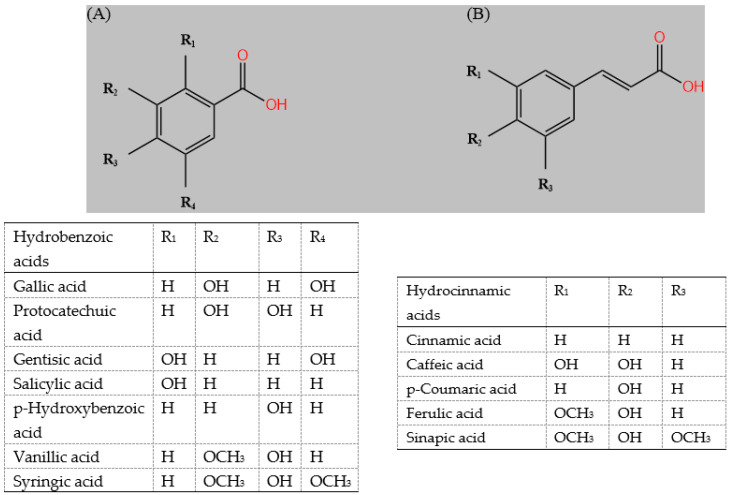
Structure of the hydrobenzoic acids (**A**), hydrocinnamic acids (**B**), 3-Deoxyanthocyanidins (**C**) and proanthocyanidins (**D**) in sorghum grains [3,6,15,17,18].

**Table 1 molecules-27-03246-t001:** Summarization of the changes of bioactive compounds and biological activities during different processing.

Processing	TPC	TFC	CTC	TAC	3-DXAs	VE	β-Carotene	Phytate	AA	HA
Decortication	↓	↓							↓	
Soaking	↓ or ↑	↓ or ―	↓ or ―				↓		↓ or ―	↓
Steaming	↓ or ↑	↓ or ―	↓ or ―						↓ or ―	
Boiling	↓	↓	↓	↓				↓	↓	
Roasting	↑ or ―	↓ or ↑	↑		―	↑	↓		― or ↑	↑
Oven	―		↓		↓ or ―	↑	↓		↓ or ―	
Microwave oven	―				―	↑	↓		―	
Extrusion	↓ or ↑		↓		―	↓	↓		↑ or ↓	
Germination	↑ or ―	↓			↑				↑ or ↓	
Fermentation	↓ or ↑	↑	↓						↑ or ↓	↑
Enzymatic treatment	↓		↓							

Note: TPC: total phenolic content; TFC: total flavonoid content; CTC: condensed tannin content; TAC: total anthocyanin content; 3-DXAs: 3-deoxyanthocyanidins; VE: vitamin E; AA: antioxidant activity; HA: hypoglycemic activity; ↑: increase; ↓: decrease; ―: not affected.

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
