# Peer review of "The Effects of Processing on Bioactive Compounds and Biological Activities of Sorghum Grains"

_molecules, 2022, doi:10.3390/molecules27103246_

Round 1

Reviewer 1 Report

This review is well constructed, of value for the scientific audience, I have a few suggestions for authors as follows:

1- Authors are encouraged to add a figure to summarize their main conclusions of this review, as the provided tables are very crowded.

2- Tables are very crowded, please try to summarize.

3- Please provide a geographical preference for certain processing methods of sorghum.

4- Detailed effects of different processing methods on sorghum chemical constituents should be provided.

Author Response

Dear Reviewer,

I greatly appreciate both your help and that of the referees concerning improvement to this paper. I hope that the revised manuscript is now suitable for publication. The revised parts are marked with red in the revised manuscript. Here below is our description on revision according to your comments.

Comment 1: Authors are encouraged to add a figure to summarize their main conclusions of this review, as the provided tables are very crowded.

Response: We added the relevant information.

Comment 2:  Tables are very crowded, please try to summarize.
Response: Thanks your advice. We removed original tables and rewritten a table.

Comment 3: Please provide a geographical preference for certain processing methods of sorghum.

Response: Thanks your advice. We added the relevant information.

Comment 4: Detailed effects of different processing methods on sorghum chemical constituents should be provided.

Response: Thanks your advice. We added the relevant information in Table 1.

Once again, thank you very much for your comments and suggestions.

Wish best wishes,

Yours sincerely,

ZhenHua Li

Reviewer 2 Report

This is a rather brief and general review on the effects of various processing operations on
bioactive constituents and bioactivities of sorghum grains, neither critical nor
comprehensive. Other than the facts and findings collected directly from the literature, the
review provides few independent views or new insights from the authors. The following
comments may be useful to improve the Ms.
For the Molecules journal, it is essential to provide more information about the effects of
processing on the molecular structures and properties of the bioactive compounds.
Add a section on the chemical composition, major bioactive compounds and quantities in
the sorghum grains with a summary table.
Add a section on the common processing methods and operating conditions in the industry
and the new methods with a summary table.
In addition to describe the effects of processing methods on the bioactivities found in the
literature, it is important to tell how to minimize the negative effects by selecting more
favourable operating conditions and alternative processing methods.
In most of the Tables, it is simply a list the effects of various operations from different
references, but not well organized to show or summarize the general trends.
Technically loose statements:
L9: “Sorghum...has high levels of bioactive compounds compared to other cereals”: This
may not be true for all bioactive compounds and compared with all cereal species.
L10: “Food processing can modify...then change biological activities of sorghum grains.”:
What are the definitions of functional, and nutritional properties and biological activities
respectively, and what are their differences?
L45: “However, bBioactive compounds in cereal grains are easier to dependent on be
affected by several factors... [9]. Some bioactive compounds are also very sensitive when in
exposure to light, heat and oxygen.”
Is soaking a biological method?

Author Response

Dear Reviewer,

I greatly appreciate both your help and that of the referees concerning improvement to this paper. I hope that the revised manuscript is now suitable for publication. The revised parts are marked with red in the revised manuscript. Here below is our description on revision according toyou’re your comments.

Comment 1: For the Molecules journal, it is essential to provide more information about the effects of processing on the molecular structures and properties of the bioactive compounds.

Response:  Thanks your advice. We added the relevant information.

Comment 2: Add a section on the chemical composition, major bioactive compounds and quantities in the sorghum grains with a summary table.

Response:  Thanks your advice. We added the relevant information.

Comment 3: Add a section on the common processing methods and operating conditions in the industry and the new methods with a summary table.

Response:  Thanks your advice. We added a figure to explain grain processing.

Comment 4: In addition to describe the effects of processing methods on the bioactivities found in the literature, it is important to tell how to minimize the negative effects by selecting more

favourable operating conditions and alternative processing methods.

Response:  Thanks your advice. We added the relevant information.

Comment 5: In most of the Tables, it is simply a list the effects of various operations from different

references, but not well organized to show or summarize the general trends.

Response:  Thanks your advice. Thanks your advice. We removed original tables and rewritten a table.

Comment 6: L9: “Sorghum...has high levels of bioactive compounds compared to other cereals”: This may not be true for all bioactive compounds and compared with all cereal species.

‐ L10: “Food processing can modify...then change biological activities of sorghum grains.”:

What are the definitions of functional, and nutritional properties and biological activities

respectively, and what are their differences?

‐ L45‐: “However, bBioactive compounds in cereal grains are easier to dependent on be

affected by several factors... [9]. Some bioactive compounds are also very sensitive when in

exposure to light, heat and oxygen.”

Response: Thanks your advice. We rewritten the relevant information.

Once again, thank you very much for your comments and suggestions.

Wish best wishes,

Yours sincerely,

ZhenHua Li

Round 2

Reviewer 1 Report

The authors responded properly to all of my comments, and the manuscript is much improved.

Author Response

Dear Reviewer,

Thank you very much for your comments and suggestions again.

Wish best wishes,

Yours sincerely,

ZhenHua Li

Reviewer 2 Report

Major revision still needed reconsideration. Authors are requested to revise and improve the whole Ms more seriously before resubmission. Following are only some of the problems:

Although a new section “1. Bioactive compounds in sorghum” has been added in response to my previous comment, but the quantities or contents of major bioactive compounds are not provided. The section number is also incorrect.

L12: “analysis”: (1) grammar mistake; (2) the review is not exactly an “analysis” of…. 

L73: “Table 1 Summarization of the changes of bioactive Compounds and biological activities during different processing”: language problems with “Summarization” and “during different processing”.

L364: “Food processing has significant and important effects on...”: what are the differences between “significant” and “important” effects?

L384-390: Most of the statements in this section are irrelevant to the review topic or meaningless to the readers, plus numerous language errors. 

Author Response

Dear Reviewer,

I hope that the revised manuscript is now suitable for publication. The revised parts are marked with red in the revised manuscript. We polished the language and modified the section number of our manuscript as you suggested. We cannot list the detailed information major bioactive compounds in sorghum grains, because several informative reviews depicted the bioactive compounds and bio-logical activities in sorghum grains, and the quantities or contents of major bioactive compounds have been provided in our published paper “Bioactive Compounds and Biological Activities of Sorghum Grains”.

Once again, thank you very much for your comments and suggestions.

Wish best wishes,

Yours sincerely,

ZhenHua Li